# Serological evidence for the presence of wobbly possum disease virus in Australia

**Anita Tolpinrud**[1,2]*, **Simon M. Firestone**[1], **Andrés Diaz-Méndez**[1], **Leanne Wicker**[2], **Stacey E. Lynch**[3], **Magdalena Dunowska**[4◎], **Joanne M. Devlin**[1◎]

**1** Asia Pacific Centre for Animal Health, Melbourne Veterinary School, The University of Melbourne, Parkville, Victoria, Australia, **2** Australian Wildlife Health Centre, Healesville Sanctuary, Zoos Victoria, Badger Creek, Victoria, Australia, **3** Agriculture Victoria Research, AgriBio Centre for AgriBioscience, Bundoora, Victoria, Australia, **4** School of Veterinary Science, Massey University, Palmerston North, New Zealand

◎ These authors contributed equally to this work.
* anita.tolpinrud@unimelb.edu.au

**Data Availability Statement:** All relevant data are within the manuscript and its Supporting Information files.

**Funding:** This work was supported by the Massey University Research Fund (grant number 20774)

## Abstract

Wobbly possum disease virus (WPDV) is an arterivirus that was originally identified in common brushtail possums (*Trichosurus vulpecula*) in New Zealand, where it causes severe neurological disease. In this study, serum samples (n = 188) from Australian common brushtail, mountain brushtail (*Trichosurus cunninghami*) and common ringtail (*Pseudocheirus peregrinus*) possums were tested for antibodies to WPDV using ELISA. Antibodies to WPDV were detected in possums from all three species that were sampled in the states of Victoria and South Australia. Overall, 16% (30/188; 95% CI 11.0–22.0) of possums were seropositive for WPDV and 11.7% (22/188; 95% CI 7.5–17.2) were equivocal. The frequency of WPDV antibody detection was the highest in possums from the two brushtail species. This is the first reported serological evidence of infection with WPDV, or an antigenically similar virus, in Australian possums, and the first study to find antibodies in species other than common brushtail possums. Attempts to detect viral RNA in spleens by PCR were unsuccessful. Further research is needed to characterise the virus in Australian possums and to determine its impact on the ecology of Australian marsupials.

## Introduction

Wobbly possum disease (WPD) virus (WPDV) causes a fatal neurological disease of the common brushtail possum *(Trichosurus vulpecula)* and is classified within the order *Nidovirales* in the family *Arteriviridae* [1, 2]. Until recently, the virus had only been confirmed in captive and free-living possum populations in New Zealand, where it has been comprehensively studied, including in experimental infection trials [3]. Clinically, the disease is characterised by early behavioural changes followed by progressive cachexia and development of neurological signs such as intentional tremors, ataxia, difficulties climbing, and occasionally presumed blindness [2–4]. A similar clinical syndrome has more recently been described in common brushtail possums in Tasmania, while another syndrome, characterised predominantly by blindness, has been observed on mainland Australia [5]. These disease syndromes in Australian possums

and Zoos Victoria (research scholarship; no specific grant number). The funders had no role in study design, data collection and analysis, decision to publish, or preparation of the manuscript.

**Competing interests:** The authors have declared that no competing interests exist.

have not yet been extensively studied, however, two divergent WPDV sequences were recently identified in archival tissue samples from three out of nine clinically affected possums originating from New South Wales [5]. The hallmark histologic lesion in WPDV-affected possums in New Zealand is the presence of variable size infiltrates of mononuclear inflammatory cells in multiple tissues including liver, spleen, kidneys, choroids and brain [4, 6]. On mainland Australia however, where blindness is predominantly observed, the pathology described in possums presumably affected by WPD is a non-suppurative inflammation primarily limited to the brain, choroids and optic tract [5]. Genetic analysis of the available WPDV genomes has demonstrated that the two newly identified Australian WPDV viruses clustered together with the New Zealand virus and were between 71 and 74% identical to each other and to the New Zealand variant over an 1,787 aa region comprising a conserved RdRp protein [5]. The existence of such diverse WPD viruses, possibly even representing separate species, is reminiscent of the situation observed for simian haemorrhagic fever arteriviruses that circulate among various non-human primates in Africa [7] and suggests that some WPDV variants may still remain undiscovered. Based on the available data, WPDV appears to have separated early in the evolution from the current members of the family *Arteriviridae* [5, 8], suggesting that it may have co-evolved with its possum host. If so, WPDV was most likely brought to New Zealand at the time when possums were introduced from their native Australia in the late 1800s [9]. Despite its likely origins, WPDV in Australian possums is not well understood. The aim of the study was to screen Australian possums for evidence of exposure to WPDV, in order to better understand the biology and epidemiology of WPDV in Australia and its distribution across different geographical regions and different possum species. This study was performed as part of a larger project that aimed to identify a range of infectious agents in Australian possums.

## Materials and methods

### Ethics

The study and sampling protocol were approved by Zoos Victoria Animal Ethics Committee (project code ZV16007) and the University of Melbourne's Faculty of Veterinary and Agricultural Sciences Animal Ethics Committee (project code #1613904.1). Sampling was performed with a Wildlife Act 1975 research permit from the Victorian Department of Environment, Land, Water and Planning (permit no. 10008226). The authors confirm that the ethical policies of the journal, as noted on the journal's author guidelines page, have been adhered to.

### Sources of samples

Samples were opportunistically collected from four sources: 1) wild common ringtail possums (*Pseudocheirus peregrinus*, 55 serum and 89 spleen samples from 99 possums) or common brushtail possums (74 serum and 81 spleen samples from 104 possums) that presented to wildlife veterinary hospitals located in the Melbourne (Victoria) area between January 2016 and April 2018; 2) spleen samples collected from euthanised or naturally deceased common ringtail (n = 14) and common brushtail (n = 7) possums that presented to veterinary hospitals in north Sydney (New South Wales) between 2016 and 2017; 3) archival serum samples from wild common brushtail (n = 25) and common ringtail (n = 2) possums that presented to the veterinary hospital at Adelaide zoo (South Australia) between April 2013 and November 2017; and 4) archival serum samples from wild caught, apparently healthy mountain brushtail possums (*Trichosurus cunninghami*, n = 31) and one common brushtail possum, collected between 2005–2007 and 2014–2017 from the Strathbogie Ranges, Victoria [10, 11]. Altogether, 188 serum samples and 191 spleen samples from 283 possums were available for testing. Where possible, information regarding age, sex and environment the possum originated from (urban,

semi-urban or rural) was recorded for each sample. For simplicity, the sampling areas were characterised by their most prominent environment. In the case of the Victorian samples, "urban" was defined as a semi-circular area bounded by Melbourne's main metropolitan ring road and encompassing the majority of the most heavily built up areas. Samples from Greater Melbourne that were collected outside this area were defined as "semi-urban", whilst those from the Strathbogie Ranges were defined as "rural". Samples from South Australia and New South Wales were collected from urban areas in Adelaide and Sydney respectively.

## Sample collection

Peripheral blood (0.5 to 2 mL) was collected into plain blood collection tubes under isoflurane general anaesthesia. Serum was separated from the clotted blood by low-speed centrifugation and stored at either -20 ˚C or -80 ˚C until testing. Aliquots were shipped to Massey University for WPDV antibody detection using ELISA. Spleen samples were collected at necropsy from fresh, chilled or previously frozen carcasses of possums that were either euthanised or had died for reasons unrelated to this study. The samples were placed in sterile 2 mL microcentrifuge tubes and frozen at -80 ˚C. Small (less than 0.5 mm in each dimension) pieces of spleen were later dissected from each sample after thawing, submerged in 10 x volume of RNA*later*® solution (Invitrogen, USA) and shipped on ice blocks to Massey University for WPDV specific reverse transcriptase quantitative PCR (RT-qPCR) testing.

## WPDV serology

Serum samples were tested for the presence of WPDV antibodies using an indirect ELISA based on a recombinant nucleocapsid (rN) protein of the virus. Sera were diluted 1:10 in phosphate buffered saline pH 7.5 containing 0.05% (v/v) Tween 20 (PBST) and tested in duplicate according to the protocol described previously [12], with the exception that substrate development was stopped after 3–5 minutes, based on the strength of the colour in positive control wells. The results were presented as corrected optical density at 450 nm ($OD_{450}$). Previously established cut-off values were used to classify samples as negative ($OD_{450} < 0.28$), suspect or equivocal ($OD_{450} = 0.28$–0.41) or positive ($OD_{450} > 0.41$) for WPDV antibodies [12]. As the rN antigen used in the blocking ELISA test was based on the sequence of the New Zealand WPDV, the predicted amino acid sequences of nucleocapsid (N) protein from the newly available two Australian WPD viruses were aligned with that of the New Zealand WPDV. The multiple alignments were performed using Clustal Omega (version 1.2.4) with default settings (available at https://www.ebi.ac.uk/Tools/msa/clustalo/).

## WPDV specific qRT-PCR

Viral RNA was extracted from approximately 10 mg of spleen using the NucleoMag® Vet kit (MACHEREY-NAGEL, Germany) on a KingFisher™ Flex Purification System, according to the manufacturer's instructions, and eluted with 100 μL of the supplied elution buffer. Complementary DNA (cDNA) was made using 2 μL qScript™ Supermix (Quanta Biosciences, USA) and 8 μL RNA in a 10 μL reaction, according to the manufacturer's instructions. Spleen samples (n = 191) were tested in duplicate on a Mic qPCR instrument (Bio Molecular Systems), using 1 μL template and PowerUp™ SYBR® Green Master Mix (Applied Biosystems, USA), with the primer concentration and cycling conditions as described previously [13]. Primers targeted the RNA dependent RNA polymerase gene within ORF1b [13]. Samples were considered positive if the amplification curve crossed the automatically defined threshold and the melting peak was between 85 ˚C and 86.5 ˚C. Samples were considered equivocal if only one of the duplicates was positive with the Cq value >33 or if both duplicates showed the correct

melting peak, but the Cq was >37. Samples with equivocal results were retested in duplicate with 2 μL and 5 μL template, using a conventional PCR targeting a 321 bp conserved region in ORF1b [14].

## Statistical analysis

Statistical analysis was carried out using Stata/SE 14.2 (StataCorp, USA). Frequency of WPDV antibody detection was calculated as the proportion of seropositive animals in the study population, with exact 95% confidence intervals (CI). Fisher's exact test was used to test for differences in seroprevalence by species, sex, age and environment. The analysis was run twice, using both the higher (> 0.41) and lower (> 0.28) ELISA cut-off values for sample positivity. Adjustment for test imperfection to estimate true prevalence was performed using the Rogan-Gladen estimator, based on previously determined estimates for test sensitivity and specificity [12, 15].

## Results

### Serology

Approximately half of the serum samples came from common brushtail possums. Adults, males and possums from urban areas were over-represented in the sampled population (Table 1). Thirty out of 188 serum samples (16.0%, 95% CI 11.0–22.0) tested positive for WPDV antibodies, while a further 22/188 (11.7%, 95% CI 7.5–17.2) were equivocal. The remaining 136 serum samples were negative for WPDV antibodies. Adjusting for test imperfection, the true seroprevalence of WPDV in this sample set would be 22.0% (95% CI 14.2–32.0%). Seropositive, equivocal and seronegative individuals were identified amongst both the recently collected and archival samples, including the mountain brushtail samples from 2006 (2/14 seropositive and 2/14 equivocal). The percentage of WPDV seropositive possums

**Table 1. Seroprevalence results and statistical analysis assessing individual epidemiological variables for wobbly possum disease virus seropositivity in a sample of Australian possums (n = 188), using the cut-off value of corrected $OD_{450}$ > 0.41 for positive samples.**

|  | WPDV seropositive | Prev. (%) | 95% CI | p value* |
|---|---|---|---|---|
| **Species** |  |  |  | 0.38 |
| Common ringtail | 6/57 | 10.5 | 4.0–21.5 |  |
| Mountain brushtail | 5/31 | 16.1 | 5.5–33.7 |  |
| Common brushtail | 19/100 | 19.0 | 11.8–28.1 |  |
| **Sex** |  |  |  | 0.84 |
| Male | 17/106 | 16.0 | 9.6–24.4 |  |
| Female | 13/75 | 17.3 | 9.6–27.8 |  |
| Unknown | 0/7 | 0.0 | 0.0–41.0 |  |
| **Age** |  |  |  | 0.16 |
| Juvenile | 5/32 | 15.6 | 5.3–32.8) |  |
| Subadult | 1/26 | 3.9 | 0.1–19.6 |  |
| Adult | 24/126 | 19.1 | 12.6–27.0 |  |
| Unknown | 0/4 | 0.0 | 0.0–60.2 |  |
| **Environment** |  |  |  | 1.00 |
| Rural | 5/32 | 15.6 | 5.3–32.8 |  |
| Urban | 15/91 | 16.5 | 9.5–25.7 |  |
| Semi-urban | 10/65 | 15.4 | 7.6–26.5 |  |

* calculated using Fisher's exact test, excluding the unknown groups

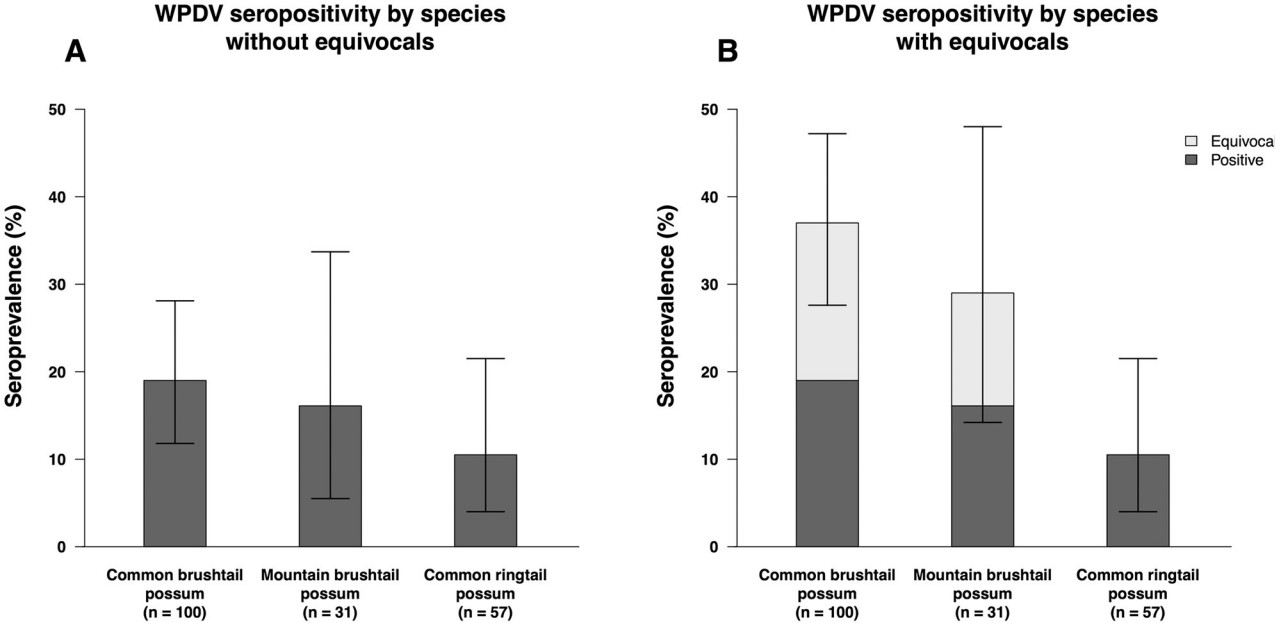

**Fig 1. The percentage of possums seropositive for antibodies to WPDV in Australia, stratified by species.** Samples were considered positive if their corrected $OD_{450} > 0.41$ and equivocal if $0.28 <$ corrected $OD_{450} \leq 0.41$. Sera (n = 188) were obtained from wild-caught, apparently healthy possums (archival samples) or injured/sick possums presented to wildlife centres in various geographical regions in Australia. Bar plots showing the percentage of seropositive samples in A and including both seropositive and equivocal samples in B, with 95% confidence intervals.

stratified by species and geographical area is shown in Figs 1 and 2. WPDV seropositivity was comparable between groups based on sex, age, environment and species using the higher ($> 0.41$) cut-off value for positive samples (Table 1). When the lower ($> 0.28$) cut-off value was used, common brushtail possums had higher seropositivity than possums from the other two species (p = 0.001), and the WPDV seropositivity was higher after possums reached adulthood (p = 0.026; S1 Table).

The comparison of predicted protein N sequences of the Australian and New Zealand WPD viruses showed that the Australian aa sequences were about 70% identical to the sequences from the New Zealand virus (Fig 3).

## WPD RT-qPCR

A total of 191 spleen samples were tested for WPDV using RT-qPCR. Of these, 35 samples were considered equivocal. However, all of these were negative when tested with the conventional PCR, and hence all 191 samples tested were considered negative for WPDV RNA.

## Discussion

This study is the first to demonstrate antibodies to a WPDV, or an antigenically similar virus, in Australian possums. At least 16.0% of sampled possums had WPDV antibody using the rN ELISA test. The use of rN protein as antigen in this ELISA was informed by previous studies showing this protein to be immunodominant in other arteriviruses [16–18]. The immunogenicity of this protein in WPD has been demonstrated in experimentally infected possums in a previous study using the same ELISA method and antigen [12]. In addition, WPDV N protein has no similarity to any other proteins currently deposited in public databases based on BLAST

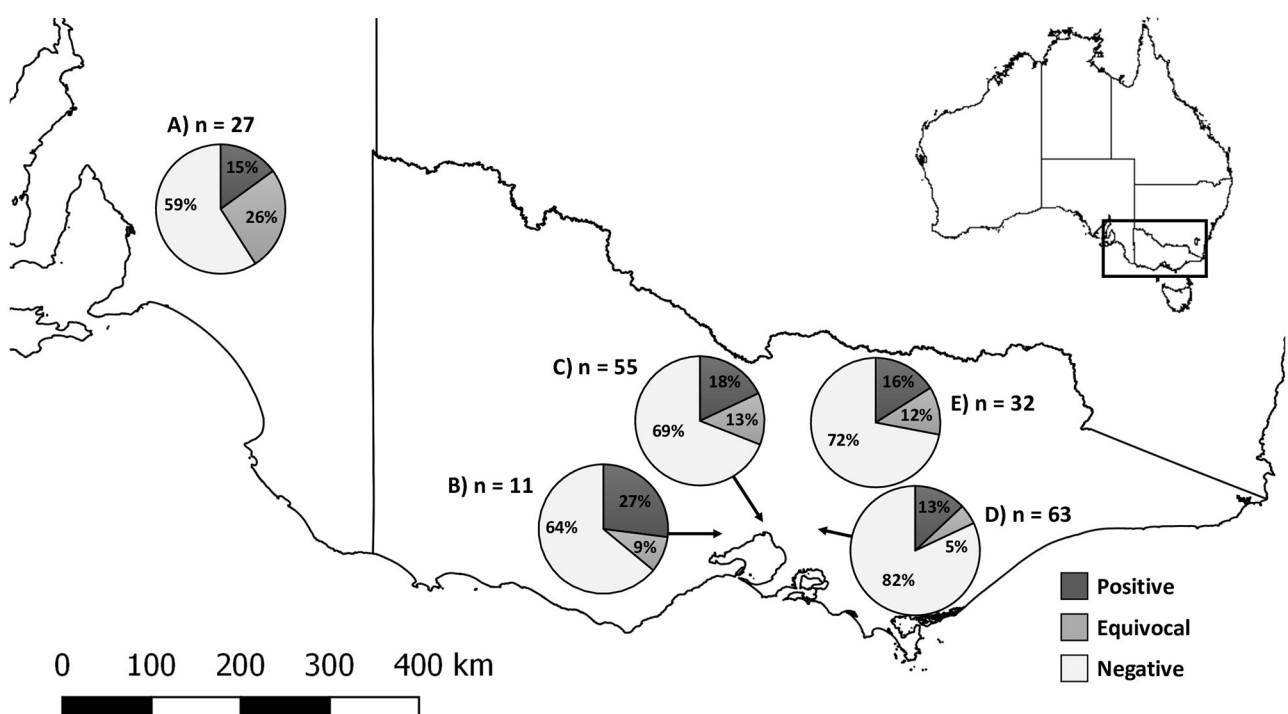

**Fig 2. Geographical distribution of sampled possums that were seropositive or suspect seropositive for WPDV in Australia.** Sera (n = 188) originated from five regions in the states of Victoria and South Australia: A- Adelaide; B—Werribee; C—Melbourne zoo; D—the eastern suburbs of Melbourne and the Yarra Valley; E—Strathbogie mountains. Seropositivity for wobbly possum disease virus (WPDV) was defined by a corrected $OD_{450}$ > 0.41 and suspect positives (equivocal) by a corrected $OD_{450}$ > 0.28. Base maps and state boundaries were sourced from GEODATA TOPO 250K Series 3 (Geoscience Australia; http://pid.geoscience.gov.au/dataset/ga/64058 accessed 28 July 2019) and reproduced under Creative Commons Attribution License 4.0.

searches. The predicted N proteins from recently characterised Australian WPD viruses were approximately 70% identical to the rN protein used as an antigen in the blocking ELISA test. This is a substantial level of identity; however, the impact these sequence differences may have on the detection of antibodies to the different viruses is not known. Considering these

```
WPDV-NZ    MAARRRSRQNGRSSSRPMARNAPPTGRTGRQRRNYITPEQRSAVTLARQYGSASSASPVA    60
WPDV-AU1   MVRRRQSRQNGRSSSRPMGRTGPPTARRGRRRQNYITPEQKSALTLARQYGSPNTSSPVN    60
WPDV-AU2   MAPRRQSRQNGQSSSRRQGRNAQPMARRGRQRRNFTTQEQRSALTLARQYGSPSTANPVN    60
            *.  **:.*****:****   .*.. *  .* **:*:*:  * **:**:*********  ...:.**

WPDV-NZ    GITGLLKPHYSPDATLVTGDAREIGPAACRLIARAALALAQGHGEIIANDEAFVFALTLP    120
WPDV-AU1   SITALLKPHFSPDASLDQDDARDAGPVACRIIARAALALAQGHGHISVSDEAFAIQLTLP    120
WPDV-AU2   AITSLLKPHFSPDASLLPDDARDAGPVACRVIARAALALTQGHGHIRVSDEAFAVELTLP    120
            .**.*****:****:*   ***. ** .***:********:*****.*  ..****..  ****

WPDV-NZ    RKGRNA--   126
WPDV-AU1   RKSRNAGQ   128
WPDV-AU2   RKSRNAGQ   128
            **.***
```

| % identity: | | | |
|---|---|---|---|
| WPDV-NZ | 100 | 71.43 | 69.84 |
| WPDV-AU1 | 71.43 | 100 | 78.91 |
| WPDV-AU2 | 69.84 | 78.91 | 100 |

**Fig 3. Multiple alignment of the predicted protein N sequences of the New Zealand and Australian Wobbly Possum Disease Viruses (WPDVs).** The alignments were done in Clustal Omega using default parameters. The regions used for the alignments comprised nucleotides 12241–12627 (WPDV-AU1, GenBank accession MN635447), 12017–12403 (WPDV-AU1, GenBank accession MN635448) and 12424–12804 (WPV-NZ, GenBank accession number JN116253. The percent identity values are also shown.

observations, our data provide strong evidence that WPDV, or antigenically similar viruses, circulate among Australian possum species.

The cut-off value of $OD_{450} > 0.41$ used for positive samples was previously shown using Bayesian latent class analysis [12] to have a sensitivity of 0.62 and specificity of 0.97, while the sensitivity and specificity using the lower cut-off ($OD_{450} > 0.28$) was 0.79 and 0.84, respectively. The establishment of the cut-offs was hindered by the lack of known negative sera [12]. In this study, we initially elected to use the more conservative, higher cut-off with a higher specificity and less likelihood of false positive results. However, due to the uncertainty surrounding these cut-off values, the potential for misclassification bias needs to be considered when interpreting the results and, consequently, the results in this study were re-analysed using the lower cut-off for completion. Nonetheless, even after adjusting for test imperfection and using the most conservative cut-off, our estimate of the true seroprevalence of WPDV (22.0%; 95% CI 14.2–32.0%) provides strong evidence that the virus is present and circulating in the country. The detection of seropositive individuals from regional Victoria in 2006 would support this finding. The potential degradation of sample quality related to long-term storage and repeated freeze-thaw cycles of historical samples is worthy of consideration, particularly with regards to the equivocal results for some samples; however, other studies have demonstrated that antibodies are relatively robust to such changes [19–21].

The proportion of seropositive possums in Australia (16.0% positive and 11.7% equivocal) was similar to that reported in New Zealand (20.9% positive and 11.7% equivocal) using the same ELISA [12]. This is unexpected, considering that New Zealand and Australian possums live in very different environments, with New Zealand possums (which are considered pests [22]) existing in large numbers [23] and in high densities, which could facilitate the transmission of WPDV [3]. The density of possums in the Australian sampling locations was presumably lower, but possum home range can increase with the decrease in possum density [24], potentially facilitating similar levels of contact between possums in populations of various densities, thereby contributing to a similar proportion of seropositive animals in New Zealand and Australian-based studies. Previous research and infection studies have suggested that WPDV can be transmitted through direct or indirect contact, either with infected individuals or surfaces contaminated with body secretions from infected animals, although the exact routes of natural transmission have not been fully elucidated [3, 25]. Documented modes of transmission of other arteriviruses include contact with infected individuals, fomites or aerosols from infective body secretions, as well as venereal and *in utero* transmission [26–28]. The possibility that WPDV could be mechanically transmitted by flying insects, prevalent in Australia, should also be considered in future studies [29, 30].

We have also shown for the first time that antibodies to WPDV, or an antigenically similar virus, are present in possums other than common brushtails, including common ringtail possums and mountain brushtail possums. Australia is home to more than 20 possum species, as well as many other marsupials [31]. Hence, it would be of value to determine the full spectrum of susceptibility to WPDV infection of various marsupial species. Such data would be useful to better understand the ecology of the virus in Australia and its clinical implications, as well as to help inform disease intervention or management strategies (such as management of captive breeding colonies or translocation risk analyses) for endangered possum species.

The design of the current study did not allow associations between WPDV infection and disease to be investigated. This was due to the opportunistic sampling strategy employed, with the majority of samples obtained from diseased or deceased possums and no opportunities for obtaining paired samples to demonstrate rising antibody titres. Collection of tissues for concurrent histological examination could be considered in future studies to look for associated pathology, but these were not available for this sample set.

WPDV RNA was not detected by PCR in this study. This could indicate that the virus was genetically divergent from the New Zealand WPDV, although both sets of primers used in this study were designed to target well conserved regions of the nidovirus genome [13, 14]. The pairwise nucleotide similarities between the two Australian WPDV sequences (WPDV AU1 and WPDV AU2) and the New Zealand WPDV sequence were approximately 71% over the entire genome, suggesting a comparatively high level of sequence divergence between different lineages of WPDV [5]. Retrospective comparison of the Australian and New Zealand WPDV sequences within the regions targeted by the PCR primers used in the current study suggested that the qPCR primers may not have been able to detect all variants of WPD viruses, as there were three (WPDV AU1) and four (WPDV AU2) mismatches within the forward primer, as well as four (WPDV AU1) and 7 (WPDV AU2) mismatches within the reverse primer. However, the conventional PCR primers would have been expected to bind to both WPDV AU1 and AU2 sequences, as there were only two (forward primer) or one (reverse primer) mismatches between the New Zealand and Australian WPDV sequences, with full complementarity in the last three nucleotides at the 3' end.

Alternatively, failure to detect WPDV RNA in the current study could be attributed to the timing of sample collection with respect to WPDV infection. In experimentally infected possums in New Zealand, high levels of WPDV have been detected in multiple organs three to four weeks following infection, including in serologically positive possums [25]. However, the persistence of virus in naturally infected possums may be different, and there may be differences between virus lineages. Poor sample quality may have also prevented detection of viral RNA in the opportunistically collected samples, particularly after multiple freeze-thaw cycles prior to shipment and RNA extraction. Furthermore, many of the samples were collected a few days after death, which would likely have affected viral RNA quality. Future attempts to detect or isolate WPDV should therefore target a range of fresh tissues, particularly those previously found to contain high concentrations of the virus (e.g. liver, brain, spleen, lymph nodes or kidneys), with careful attention to tissue preservation and handling [13]. Availability of more viral sequences from field WPDVs circulating in different regions of New Zealand and Australia would shed some light on the evolutionary pathways of the virus in these two diverse environments.

## Supporting information

**S1 Table. Seroprevalence results and statistical analysis assessing individual epidemiological variables for wobbly possum disease virus seropositivity in a sample of Australian possums (n = 188), using the cut-off value of corrected OD$_{450}$ > 0.28 for positive samples.** (DOCX)

## Acknowledgments

We thank Prof Colin Wilks for his advice, and we thank our colleagues at Zoos Victoria, Zoos South Australia, The Australian Registry of Wildlife Health, Lort Smith Animal Hospital and the University of Melbourne for their assistance with sample collection/donation. We gratefully acknowledge the technical help of Sayani Ghosh (Massey University) and Gayathri Gopakumar (Melbourne Veterinary School).

## Author Contributions

**Conceptualization:** Anita Tolpinrud, Andrés Diaz-Méndez, Leanne Wicker, Stacey E. Lynch, Joanne M. Devlin.

**Data curation:** Anita Tolpinrud, Leanne Wicker, Magdalena Dunowska.

**Formal analysis:** Anita Tolpinrud, Simon M. Firestone.

**Funding acquisition:** Magdalena Dunowska.

**Investigation:** Magdalena Dunowska.

**Methodology:** Anita Tolpinrud, Simon M. Firestone, Magdalena Dunowska, Joanne M. Devlin.

**Project administration:** Anita Tolpinrud.

**Resources:** Leanne Wicker, Magdalena Dunowska.

**Supervision:** Andrés Diaz-Méndez, Leanne Wicker, Stacey E. Lynch, Joanne M. Devlin.

**Writing – original draft:** Anita Tolpinrud.

**Writing – review & editing:** Simon M. Firestone, Andrés Diaz-Méndez, Stacey E. Lynch, Magdalena Dunowska, Joanne M. Devlin.

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
