## [Decision Letter · Decision Letter 0]

24 Mar 2020

PONE-D-20-02745

Serological evidence for the presence of wobbly possum disease virus in Australia

PLOS ONE

Dear Dr Tolpinrud,

Thank you for submitting your manuscript to PLOS ONE. After careful consideration, we feel that it has merit but does not fully meet PLOS ONE’s publication criteria as it currently stands. Therefore, we invite you to submit a revised version of the manuscript that addresses the points raised during the review process.

We would appreciate receiving your revised manuscript by May 4, 2020. To enhance the reproducibility of your results, we recommend that if applicable you deposit your laboratory protocols in protocols.io, where a protocol can be assigned its own identifier (DOI) such that it can be cited independently in the future. For instructions see: http://journals.plos.org/plosone/s/submission-guidelines#loc-laboratory-protocols

We look forward to receiving your revised manuscript.

Kind regards,

Michelle L. Baker, PhD

Academic Editor

PLOS ONE

Journal Requirements:

2.  We note that [Figure 2] in your submission contain [map/satellite] images which may be copyrighted. All PLOS content is published under the Creative Commons Attribution License (CC BY 4.0), which means that the manuscript, images, and Supporting Information files will be freely available online, and any third party is permitted to access, download, copy, distribute, and use these materials in any way, even commercially, with proper attribution. For these reasons, we cannot publish previously copyrighted maps or satellite images created using proprietary data, such as Google software (Google Maps, Street View, and Earth). For more information, see our copyright guidelines: http://journals.plos.org/plosone/s/licenses-and-copyright.

1.     You may seek permission from the original copyright holder of Figure [2] to publish the content specifically under the CC BY 4.0 license.  

Reviewers' comments:

Reviewer's Responses to Questions

**Comments to the Author**

1. Is the manuscript technically sound, and do the data support the conclusions?

Reviewer #1: Yes

Reviewer #2: Partly

2. Has the statistical analysis been performed appropriately and rigorously? 

Reviewer #1: Yes

Reviewer #2: No

3. Have the authors made all data underlying the findings in their manuscript fully available?

Reviewer #1: Yes

Reviewer #2: Yes

4. Is the manuscript presented in an intelligible fashion and written in standard English?

Reviewer #1: Yes

Reviewer #2: Yes

5. Review Comments to the Author

Reviewer #1: Recommendation’s: The paper describes a serological study undertaken for the surveillance of arteriviruses in three species of small Australian mammals in two southern states of Australia.

The manuscript is concise, organised, and contributes to the knowledge of a long-standing health problem in the common brushtail possum, but is most relevant to an Australasian, or wildlife disease readership. The manuscript adds to our knowledgebase of the geographic distribution, host exposure, and incidence of the group of viruses associated with the wobbly possum disease syndrome. Although a very publishable manuscript, I would tend towards recommending a rejection while providing encouragement to submit to journals with a more targeted readership.

The manuscript is technically sound and the data support the conclusions. Nonetheless, the authors make it clear that another paper has been published describing the disease syndrome in Australia (citation below). The findings of the recent publication on the topic have not incorporated in the submitted manuscript introduction and are not fully woven into the discussion. This approach leaves the reader quite isolated and not updated on the current breadth of knowledge on the disease syndrome either in the NZ nor the Australian context. A more detailed introduction summarising the similarities and differences between Australian and New Zealand wobbly possum disease, including concise descriptions of the temporal, geographic range s, putative aetiologic agents identified and comparative pathology would ensure readers have a wholistic understanding of the context for the investigation described.

Recent citation:

Chang, W., Eden, J., Hartley, W.J. et al. Metagenomic discovery and co-infection of diverse wobbly possum disease viruses and a novel hepacivirus in Australian brushtail possums. One Health Outlook 1, 5 (2019). https://doi.org/10.1186/s42522-019-0006-x

---

Line 46 – summarising the histopathology – should include choroid and optic tract, should indicate that the histological lesions vary regionally (CNS only on mainland Australia, and multisystemic LP inflammation in NZ and Tas)

Line 50 – The sequence of emergent literature is not highly relevant to the reader. The readership would benefit more from an understanding of the current knowledge of the variation across the genomes of the New Zealand and two Australian arteriviruses associated with the disease syndrome described. Understanding the diversity across the three arteriviruses found within wobbly possums seems to be an important basis for interpreting serological responses. A brief summary of the temporal, geographic, and demographic nature of the syndrome in Australia would also benefit the reader and introduce the value of the serological study described.

Line 67 delete different

Line 90, perhaps euthanased or died for reasons unrelated to this study (delete naturally from various causes)

Line 144 Figure 1 and 2 rather than Fig – within the text

Line 164-166 better placed in the materials and methods

Line 170 and elsewhere – univariate seems more universally applied term rather than univariable

Line 186 – However it is currently unknown which proteins provide the main targets – could be worded in a more natural manner

Line 188-89 – May sound more natural by altering the position of the final phrase. Nonetheless, experimentally infected possums showed an increase in corrected OD450 values, using the same antigen, within four weeks….

Line 182 – first line of the discussion describes WPDV as a single entity, when three arteriviruses have been associated with the syndrome and this is important to understand when interpreting likely antigenic similarities and cross reactivity for a test designed for the New Zealand genotype - 70+% genomic divergence

Line 194-196 – The sentence does not make it clear how the findings of prevalence relate to the length of time that the syndrome and disease agents have been in the country. The citations here, do not include the most recent published report by Chang et al, where the temporal distribution of the syndrome is most thoroughly described.

Line 198 – change has been to were

Line 202 – perhaps consider rewording “This is interesting” as it seems a bit presumptive to assume what the reader may find interesting

Line 222-224 – Although samples were collected opportunistically, histological samples could have also been taken from the dead animals examined. Lymphoplasmocytic inflammation often persists through freeze-thaw artefact.

Line 224 – the language seems awkward and the premise that a single titre could ever tell you when an infection occurred seems flawed

Line 230 – it is difficult to understand how the reader could interpret the PCR findings without having knowledge of the presence of the three arteriviruses associated with wobbly possum syndrome and the extent of their genomic divergence. It seems very late to be introducing the presence and diversity of the three arteriviruses in line 239-240 – when describing primers

Line 240 – delete used

Reviewer #2: Thanks for the opportunity to review this article. It was very well written and interesting, with measured and scientifically appropriate interpretation. My main comments are in regards to model structure and the Figures.

Major comments:

Granted, the statistical approach used here is far better than the usual standard of wildlife publications, which is encouraging. However, the manual backwards stepwise elimination approach is an outdated method for estimating causal effects. A minimum approach should involve each risk factor considered separately in its own model, with covariates added in consideration of causal framework.

The figures should be constructed better for effective communication. A few suggestions:

- for Figure 1, it would be helpful to the reader to present the plots unstacked and with error margins: particularly in view of the relatively small sample sizes, error margins are essential for a reader to readily identify whether or not the seroprevalence differs between the species.

- Figure 2 is unlikely to print well given its large overall size and the relatively small pie charts; and indeed the use of pie charts is to be avoided (pie charts are a poor medium for comparing outcomes across groups). While a map of source sites is good idea, it could be communicated with a small map of Australia and icons for all sample sources, including north Sydney (omit the pie charts). The comparison of the serological outcome between sample sites could then be presented as a separate figure, in a bar chart with error margins (for the same reasons as Figure 1).

Minor comments:

Lines 29-30: Provide error margins with the proportions.

Line 39: Suggest replacing ‘Australian brushtail possum’ with ‘common brushtail possum’: it’s best to use the same common name consistently throughout the article. It could also be confusing for readers not familiar with Australian and New Zealand fauna, given the geographical references involved in the sentence too.

Line 74- 79: Given the age of some of the serum samples, it would be appropriate for brief commentary in the discussion on how that may have affected results in the discussion. It would also be interesting to know amongst relatively old samples, if both seropositives and seronegatives were detected- a sentence or two in the Results would suffice.

Line 81: Indicate the criteria used to define a possum as being urban, semi-rural or rural.

Line 131-132: The approach of modelling the ‘suspect/equivocal’ results in addition to the seropositive results was unusual, but I can understand why given that cut-offs are poorly defined. Some brief, specific commentary in the discussion on the reasoning and the potential implications of misclassification bias would be appropriate. Further, a mention of the relatively small sample size and potential for Type II error influencing study findings is required.

Line 137: It would be helpful to start the Results section with an overall summary of the possum origin of the samples, separately for ELISA and PCR samples. E.g. summarise the numbers of possums by sample source, species, sex, age and environmental origin.

Line 173: Table 1: provide error margins for the prevalence estimates

Line 194-197: Estimates of true prevalence belong in the Results section, with methods for estimation outlined in the Methods section.

Line 199-200: Or more realistically, latent class analysis could be used to estimate prevalence using imperfect tests - identification of ‘known negative’ sera of a poorly understood wildlife infection is not the practical approach. This sentence should be updated accordingly.

Line 202- 211: To complement this section, a brief comment on known modes of transmission for other arteriviruses in other species would be informative to the reader.

Line 216-219: Vague commentary like this whilst referencing endangered species is not particularly helpful. How would better understanding the ecology of the virus in Australia and its clinical implications be of value, beyond academic interest? Endemic diseases are not the primary cause of species declines in Australia, so understanding the ecology of the virus and its clinical implications to inform development of interventions to target endemic disease is not a particularly effective or efficient way to aide species recovery. So for what other reason(s) might understanding the ecology of the virus and its clinical implications be helpful for endangered wildlife? Specify clearly or consider omitting this part of the text.

6. PLOS authors have the option to publish the peer review history of their article (what does this mean?). If published, this will include your full peer review and any attached files.

Reviewer #1: No

Reviewer #2: No

---

## [Author Response · Author response to Decision Letter 0]

17 May 2020

Many thanks for the careful review of our manuscript and for the constructive comments provided. A point-by-point response to the reviewers' comments has been uploaded as a separate document, as requested.

---

## [Decision Letter · Decision Letter 1]

16 Jul 2020

PONE-D-20-02745R1

Serological evidence for the presence of wobbly possum disease virus in Australia

PLOS ONE

Dear Dr. Tolpinrud,

Thank you for submitting your manuscript to PLOS ONE. After careful consideration, we feel that it has merit but does not fully meet PLOS ONE’s publication criteria as it currently stands. Therefore, we invite you to submit a revised version of the manuscript that addresses the points raised during the review process.

Specifically, reviewer 2 has provided feedback requiring minor changes to the manuscript.

We look forward to receiving your revised manuscript.

Kind regards,

Michelle L. Baker, PhD

Academic Editor

PLOS ONE

Reviewers' comments:

Reviewer's Responses to Questions

**Comments to the Author**

1. If the authors have adequately addressed your comments raised in a previous round of review and you feel that this manuscript is now acceptable for publication, you may indicate that here to bypass the “Comments to the Author” section, enter your conflict of interest statement in the “Confidential to Editor” section, and submit your "Accept" recommendation.

Reviewer #1: All comments have been addressed

Reviewer #2: All comments have been addressed

2. Is the manuscript technically sound, and do the data support the conclusions?

Reviewer #1: Yes

Reviewer #2: Yes

3. Has the statistical analysis been performed appropriately and rigorously? 

Reviewer #1: Yes

Reviewer #2: Yes

4. Have the authors made all data underlying the findings in their manuscript fully available?

Reviewer #1: Yes

Reviewer #2: Yes

5. Is the manuscript presented in an intelligible fashion and written in standard English?

Reviewer #1: Yes

Reviewer #2: Yes

6. Review Comments to the Author

Reviewer #1: Thank you for the opportunity to further review the manuscript.

The authors have done an excellent job addressing the suggestions of both reviewers. Changes have been thoughtful, clear and concise. The table titles and figures are thoughtfully enhanced. In my opinion the manuscript is acceptable for publication. I offer very minor suggestions, which include:

- Describing the results from the NSW animals - I am certain that there would be interest from the readership

- Line 85 – Wild does not need to be capitalised

- Line 100 – divided – perhaps classified or characterised might read better

- Line 236 – circulates – to circulate

- Line 241- The establishment of the cut-offs was (rather than were) hindered

- Line 267 – although the exact routes of natural transmission have (rather than has) not been fully elucidated

- Line 316 – found to contain high concentrations (rather than concentration)

Reviewer #2: A couple of minor comments:

Lines 250: “As the authors do not suggest a single optimal cut-off, it is unclear which is most appropriate for our purpose”. This statement comes across poorly, as it vaguely insinuates criticism the validation study for not dictating a one-size-fits-all “optimal cut-off”, which would be entirely inappropriate. The optimal cut-off will depend on the circumstances of the use of the test; specifically, the relative importance of sensitivity and specificity. It is most definitely up to the study authors to decide on the most appropriate cut-off a priori, and consider the ramifications of that choice in interpreting findings. A minor tweak of the wording in view of this would be appropriate.

Line 256-257 “our estimate of true seroprevalence of WPDV (22.0%; 95% CI 14.2 – 32.0%) supports the view that this virus has been present and circulating for some time”. This statement is too tenuous and ambiguous; I would recommend citing only the more specific evidence of identification of seropositive individuals amongst samples from 2006.

Line 298: matching could be used as part of the design of an investigation into associations between the infection and disease, but it is not essential and so does not deserve specific attention here.

7. PLOS authors have the option to publish the peer review history of their article (what does this mean?). If published, this will include your full peer review and any attached files.

Reviewer #1: No

Reviewer #2: No

---

## [Author Response · Author response to Decision Letter 1]

16 Jul 2020

Reviewer #1: 

The authors have done an excellent job addressing the suggestions of both reviewers. Changes have been thoughtful, clear and concise. The table titles and figures are thoughtfully enhanced. In my opinion the manuscript is acceptable for publication. I offer very minor suggestions, which include:

Describing the results from the NSW animals - I am certain that there would be interest from the readership

Authors’ response: Unfortunately only spleen samples were available from NSW (line 89-91). As all of the spleen samples tested in this study were negative for WPDV RNA, the NSW samples are covered under the WPD RT-qPCR results (line 218-221).

Line 85 – Wild does not need to be capitalised

Line 100 – divided – perhaps classified or characterised might read better

Line 236 – circulates – to circulate

Line 241- The establishment of the cut-offs was (rather than were) hindered

Line 267 – although the exact routes of natural transmission have (rather than has) not been fully elucidated

Line 316 – found to contain high concentrations (rather than concentration)

Authors’ response: Thank you for these corrections – we have updated the manuscript to include all of these changes. 

Reviewer #2: 

Lines 250: “As the authors do not suggest a single optimal cut-off, it is unclear which is most appropriate for our purpose”. This statement comes across poorly, as it vaguely insinuates criticism the validation study for not dictating a one-size-fits-all “optimal cut-off”, which would be entirely inappropriate. The optimal cut-off will depend on the circumstances of the use of the test; specifically, the relative importance of sensitivity and specificity. It is most definitely up to the study authors to decide on the most appropriate cut-off a priori, and consider the ramifications of that choice in interpreting findings. A minor tweak of the wording in view of this would be appropriate.

Authors’ response: The sentence “As the authors of that prior study do not suggest a single optimal cut-off, it is unclear which is most appropriate for our purpose” has been changed to “In this study, we initially elected to use the more conservative, higher cut-off with a higher specificity and less likelihood of false positive results. However, due to the uncertainty surrounding these cut-off values, the potential for misclassification bias needs to be considered when interpreting the results and, consequently, the results in this study were re-analysed using the lower cut-off for completion.”

Line 256-257 “our estimate of true seroprevalence of WPDV (22.0%; 95% CI 14.2 – 32.0%) supports the view that this virus has been present and circulating for some time”. This statement is too tenuous and ambiguous; I would recommend citing only the more specific evidence of identification of seropositive individuals amongst samples from 2006.

Authors’ response: We have removed the speculative suggestion that it has been circulating “for some time” and the sentence has been changed to: “our estimate of the true seroprevalence of WPDV (22.0%; 95% CI 14.2–32.0%) provides strong evidence that the virus is present and circulating in the country”. 

Line 298: matching could be used as part of the design of an investigation into associations between the infection and disease, but it is not essential and so does not deserve specific attention here.

Authors’ response: We have reworded the sentence slightly to remove the mention of matching and the paragraph now reads: “The design of the current study did not allow associations between WPDV infection and disease to be investigated. This was due to the opportunistic sampling strategy employed, with the majority of samples obtained from diseased or deceased possums and no opportunities for obtaining paired samples to demonstrate rising antibody titres. Collection of tissues for concurrent histological examination could be considered in future studies to look for associated pathology, but these were not available for this sample set”.

---

## [Editor Report · Decision Letter 2]

21 Jul 2020

Serological evidence for the presence of wobbly possum disease virus in Australia

PONE-D-20-02745R2

Dear Dr. Tolpinrud,

We’re pleased to inform you that your manuscript has been judged scientifically suitable for publication and will be formally accepted for publication once it meets all outstanding technical requirements.

Kind regards,

Michelle L. Baker, PhD

Academic Editor

PLOS ONE

---

## [Editor Report · Acceptance letter]

24 Jul 2020

PONE-D-20-02745R2 

Serological evidence for the presence of wobbly possum disease virus in Australia 

Dear Dr. Tolpinrud:

I'm pleased to inform you that your manuscript has been deemed suitable for publication in PLOS ONE. Congratulations! Your manuscript is now with our production department. 

Kind regards, 

on behalf of

Dr. Michelle L. Baker 

Academic Editor

PLOS ONE